# Enhanced N-Type Bismuth-Telluride-Based Thermoelectric Fibers via Thermal Drawing and Bridgman Annealing

**DOI:** 10.3390/ma15155331

**Published:** 2022-08-03

**Authors:** Min Sun, Pengyu Zhang, Qingmin Li, Guowu Tang, Ting Zhang, Dongdan Chen, Qi Qian

**Affiliations:** 1State Key Laboratory of Luminescent Materials and Devices, Guangdong Engineering Technology Research and Development Center of Special Optical Fiber Materials and Devices, Guangdong Provincial Key Laboratory of Fiber Laser Materials and Applied Techniques, School of Materials Science and Engineering, South China University of Technology, Guangzhou 510640, China; s452936140@163.com (M.S.); pengyu.zhang.work@outlook.com (P.Z.); liqingmin2022@163.com (Q.L.); guowutang@126.com (G.T.); 2Nanjing Institute of Future Energy System, Nanjing 211135, China; zhangting@iet.cn; 3China Electronics Technology Group Corporation, Shijiazhuang 050051, China; 4School of Physics and Optoelectronic Engineering, Guangdong University of Technology, Guangzhou 510006, China; 5Institute of Engineering Thermophysics, Innovation Academy for Light-Duty Gas Turbine, Chinese Academy of Sciences, Beijing 100190, China

**Keywords:** n-type Bi_2_Te_3_, thermoelectric fibers, thermal drawing, Bridgman annealing

## Abstract

N-type bismuth telluride (Bi_2_Te_3_) based thermoelectric (TE) fibers were fabricated by thermal drawing and Bridgman annealing, and the influence of Bridgman annealing on the TE properties of n-type Bi_2_Te_3_-based TE fibers was studied. The Bridgman annealing enhanced the electrical conductivity and Seebeck coefficient because of increasing crystalline orientation and decreasing detrimental elemental enrichment. The TE performance of n-type Bi_2_Te_3_-based TE fibers was improved significantly by enhancing the power factor. Hence the power factor increased from 0.14 to 0.93 mW/mK^2^, and the figure-of-merit value is from 0.11 to 0.43 at ~300 K, respectively.

## 1. Introduction

The capacities of recycling heat for electricity generation or converting electricity for refrigeration have enabled the TE technique to be a promising way to recover the waste heat [1,2], a zero-carbon and ubiquitous resource. The efficiency of TE materials is gauged by its figure-of-merit value, *ZT = S^2^σT/κ*, where *S* is Seebeck coefficient, *σ* is electrical conductivity, *T* is absolute temperature, and *κ* is thermal conductivity. Layer-structure Bi_2_Te_3_-based materials have been reported to be the best room-temperature TE materials [3,4]. However, the commercial n-type Bi_2_Te_3_-based materials possess a *ZT* restricted to <1 at room temperature, owing to the common interdependence of electron-phonon transport and more sensitivity to texturing than their p-type counterparts [5,6]. Moreover, p-type and n-type Bi_2_Te_3_-based materials mostly compose TE devices together in block shapes, which are based on rigid design, so they cannot be used on heat sources with irregular shapes [7,8]. Current solutions focus on mainly embedding/coating the Bi_2_Te_3_-based materials into/on flexible substrates such as yarns, textiles, and papers [9,10,11]. But many drawbacks remain, such as low *ZT* and weak mechanical flexibility, which still impede their progress.

Hundred-kilometer-length silica fibers for optical communication are typically produced by thermal drawing a preform at the softening temperature of silica glass [12]. With the same facile technique, multiple materials with disparate properties, including semiconductors, metals, and insulators, can be co-drawn into a micro-nano scale [13,14]. It paves the way to multi-material fibers endowed with unique functionalities at fiber length scales and costs. In recent years, researchers have demonstrated ultralong TE fibers by the thermal drawing method to integrate crystalline TE micro/nanofibers into a glass-fiber template [15,16,17,18,19]. Although p-type Bi_2_Te_3_ fibers are reported with a high *ZT*~1.4 at room temperature [20], n-type Bi_2_Te_3_ fibers have not been systematically studied or improved *ZT* values effectively by thermal drawing or post-treatment, resulting in TE fiber devices with p-n pairs cannot be widely used. The p-n pair fibers then have the potential to be applied in the field of wearable self-powered devices and temperature-sensing fabrics [7,21,22,23].

Herein, n-type Bi_2_Te_3_-based TE fibers with mechanical flexibility were fabricated by optical-fiber-template thermal drawing and Bridgman annealing, also called the Bridgman–Stockbarger method for single-crystal growth and is recently applied in the fabrication of multi-material single-crystal fibers [24,25]. Through Bridgman annealing, the detrimental elemental enrichment in the fiber cores can be reduced, and crystalline orientation along (0 0 *l*) can be increased, so they possess enhanced electrical transport. The *ZT* value of fibers is measured to be ~0.43 at 300 K, approaching four times their as-drawn fiber counterparts.

## 2. Experimental Procedure

### 2.1. Fabrication

As shown in Figure 1a, the two-step method of thermal drawing and Bridgman annealing was applied for n-type Bi_2_Te_3_-based fibers fabrication. First, Bi_2_Te_3_ and Bi_2_Se_3_ powders (99.999% purity, Aladdin, Shanghai, China) are sealed by a ratio of 5:1 into a Pyrex3.3-borosilicate-glass tube with a 3-mm inner diameter and a 7-mm outer diameter. It is special that the preforms before drawing endured 700 ℃ for one-hour preheating time to form a wetting interface of the core and the cladding by elemental diffusion. Bi_2_Te_3_-based-core and glass-cladding fibers were then drawn at ~900 ℃ using an optical fiber drawing tower. All fibers maintain a complete core-cladding structure, and the fiber cores show a relative density of ~97%. N-type Bi_2_Te_3_ cores were produced into meters of continuous fibers.

Figure 1b shows the schematic of the Bridgman annealing technique. Only one ring-shaped resistance was used as a heater to offer a temperature gradient field, which is beneficial to maintaining the stability of the crystal growth interface [24]. The as-drawn fiber gradually descended and crossed the high-temperature zone to recrystallize the core at a constant speed of 10 mm/h, referencing the growth of single-crystal Bi_2_Te_3_ bulks. The high-temperature zone is ~60 °C higher than the melting point of the Bi_2_Te_3_ (585 °C) and lower than the softening point of the silicate cladding glass (719 °C).

### 2.2. Measurements

The Bi_2_Te_3_-based core was obtained by etching the as-drawn and annealed fibers in HF acid solution to strip the glass cladding and then identified by an X-ray diffractometer (XRD, X’Pert PROX, Cu K*α*, PANalytical Corp., Almelo, The Netherlands). Energy-dispersive X-ray spectroscopy (EDS) elemental studies were performed on the fiber cross-section by using scanning electron microscopy (SEM, Zeiss Merlin, ZEISS Corp., Oberkochen, Germany).

The Seebeck coefficients (*S*) or the electrical conductivities (*σ*) of some fibers were measured by the four-probe method [20]. For each sample, 3-time fiber measurements were carried out under the same conditions, and we used the average values of 3-time fiber measurements as the measuring value. All absolute values of these relative deviations are lower than 5% and can be negligible, showing the setup is effective, and the measuring results are reliable and reproducible. The thermal conductivity (*κ*) was measured using the time-domain thermal-reflection method, and the relative deviation was lower than 10%.

## 3. Results and Discussion

### 3.1. Microstructure

The XRD patterns of the as-drawn fibers and the Bridgman annealed fibers are shown in Figure 2. It is observed that all XRD peaks can be indexed to the Bi_2_Te_2.5_Se_0.5_ hexagonal phase (JCPDS No. 51-0643). After thermal drawing, the as-drawn fiber underwent a flash cooling process so that the fiber core would consist of polycrystals. Then the average particle size of as-drawn crystals was calculated to be ~29 nm according to the XRD peaks width using the Scherrer formula [17]. Still, the average particle size of Bridgman annealed crystals is >100 nm. There exist differences in XRD peak intensities between the as-drawn and annealed fibers. The annealed fibers show larger diffraction intensities than the as-drawn fibers at many lattice planes, including (0 0 6) and (0 0 15). These differences in diffraction peak intensities illustrate that the annealed fiber crystals would exhibit a more preferred orientation than the as-drawn fiber crystals do. Based on the Lotgering method [19], the orientation degree *F* of the (0 0 *l*) planes of polycrystals can be calculated:(1)$$F=\frac{P-P_{0}}{1-P_{0}}$$
(2)$$P_{0}=\frac{I_{0}(0\;0\;l)}{\sum I_{0}(h\;k\;l)}$$
(3)$$P=\frac{I(0\;0\;l)}{\sum I(h\;k\;l)}$$
where *P* and *P*_0_ are the ratios of the integrated intensities (*I*) of (0 0 *l*) planes to those of (*h k l*) planes for preferentially and randomly oriented samples, respectively. Hence the calculated *F* of the as-drawn fiber and annealed fiber polycrystals is 0.48 and 0.85, respectively, which means both possess a preferred orientation, and the annealed fiber exhibits a larger preferred orientation.

EDS mappings of the Bi, Te, Se, Si, and O elements on the polished cross-section samples of the as-drawn and the Bridgman annealed fibers are shown in Figure 3. For two samples, there is a little diffusion of Bi, Te, and Se from the core into the cladding region, and there is little diffusion of Si and O from the cladding into the core. Meanwhile, it is noted that there are Te enrichments in the as-drawn fiber core, but no enrichment is found in the Bridgman annealed fiber core. The Te enrichments should be reduced after a slow annealing process with a 10 mm/h recrystallization speed. As the Te enrichments might affect the electrical and thermal transport [26], the fiber cores with/without Te enrichments will exhibit different TE properties.

### 3.2. TE Properties

The Seebeck coefficients (*S*) and the electrical conductivities (*σ*) of fibers were examined by a self-built setup as shown in the previous study [25] and presented in Figure 4a,b. Both the as-drawn fiber and the Bridgman-annealed fiber exhibit a metallic resistance characteristic in Figure 4a, meaning that the *σ* decreases with the increasing temperature, 283–323 K. The *σ* of the Bridgman-annealed fiber is more than double of the as-drawn fiber at the same temperature. In Figure 4b, The *S* of the Bridgman-annealed fiber is more than 150% of the as-drawn fiber at the same temperature. The power factors (*PF* = *S*^2^*σ*) of fibers were calculated and presented in Figure 4c. The *PF* of the Bridgman-annealed fiber is more than four times the as-drawn fiber, and the highest value is obtained at 283 K, ~1 mW/mK^2^. It should be mainly derived from Bridgman annealing of reducing detrimental Te enrichment to increase *S* and larger orientation to enhance *σ.*

As the fiber-axis thermal conductivities (*κ*_||_) were measured on the fiber cross-section using the time-domain thermal-reflection (TDTR) method, the measured *σ*, *S*, *κ*_||_, and *ZT* of the two samples are listed in Table 1. It is observed that the *κ*_||_ of the as-drawn fiber is ultralow, which could be induced by enhanced phonon scattering from the low relative density of ~97% and the interface nanograins as reported in the previous study [17]. The *ZT* of the Bridgman-annealed fiber core is the highest in the table, three times larger than the as-drawn fiber. The *ZT* is also about twice as large as the reported *ZT* of Bi_2_Se_3_ fibers, while the Bridgman-annealed fiber core exhibits lower *σ* but a much higher *S* than other fibers [16,27]. The high *ZT* of the annealed fiber core comes from enhanced *PF*, which benefits from the preferential orientation and the elimination of the Te enrichments. Beyond these, the bending radius minimum (*r*) of the 200-μm-diameter annealed fibers is tested to be 2 cm, and their flexibility is estimated by the maximum bending strain (*ε* = *D*/2*r*~0.5) [28].

## 4. Conclusions

Herein, n-type Bi_2_Te_3_-based-core glass-clad fibers have been fabricated via thermal drawing and Bridgman annealing. All the polycrystalline Bi_2_Te_3_-based cores possess a preferential orientation, and the orientation factors of the as-drawn fiber and the annealed fiber are 0.48 and 0.85, respectively. The preferential orientation increases the fiber cores’ electrical and thermal conductivity. Importantly, the Te enrichments in the as-drawn fiber core are improved in the annealed fiber core, remarkably enhancing the Seebeck coefficient. Finally, the Bridgman annealed Bi_2_Te_3_-based core shows an enhanced *ZT* = 0.43, and our future work will be enhancing density and regulating the component and the microstructure of the fibers. In addition, the 200-μm-diameter annealed fibers can reversibly bend into a 2-cm curvature radius to show their flexibility. Such a proof-of-concept fiber drawing and annealing method have promising applications in fiber-based TE conversion.

## Figures and Tables

**Figure 1 materials-15-05331-f001:**
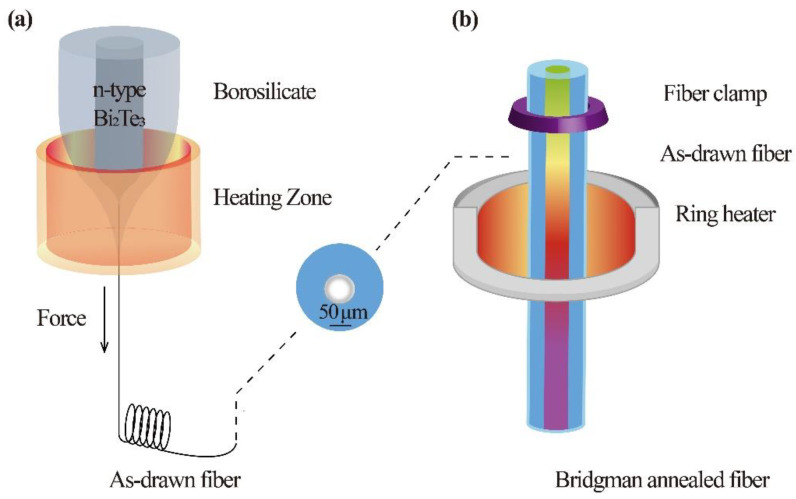
Schematic of the process of (**a**) fiber drawing and (**b**) Bridgman annealing.

**Figure 2 materials-15-05331-f002:**
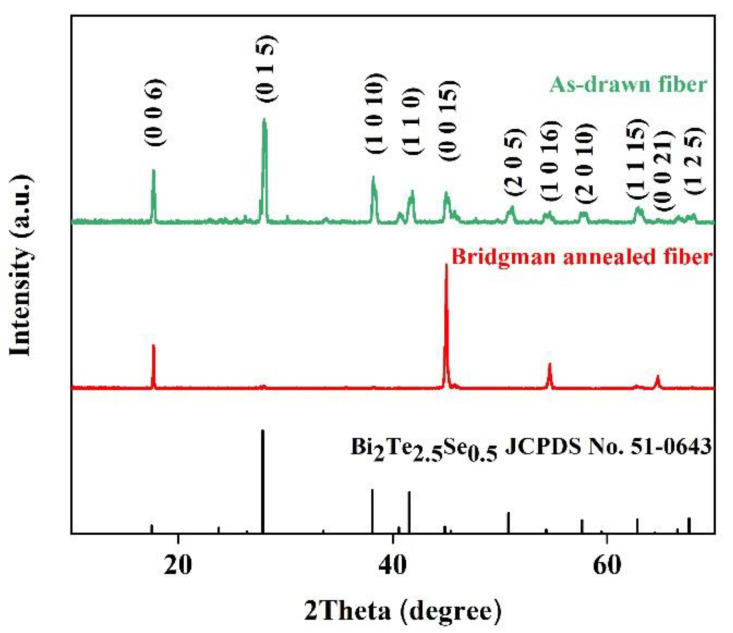
XRD patterns of the as-drawn and Bridgman annealed n-type Bi_2_Te_3_-based fiber.

**Figure 3 materials-15-05331-f003:**
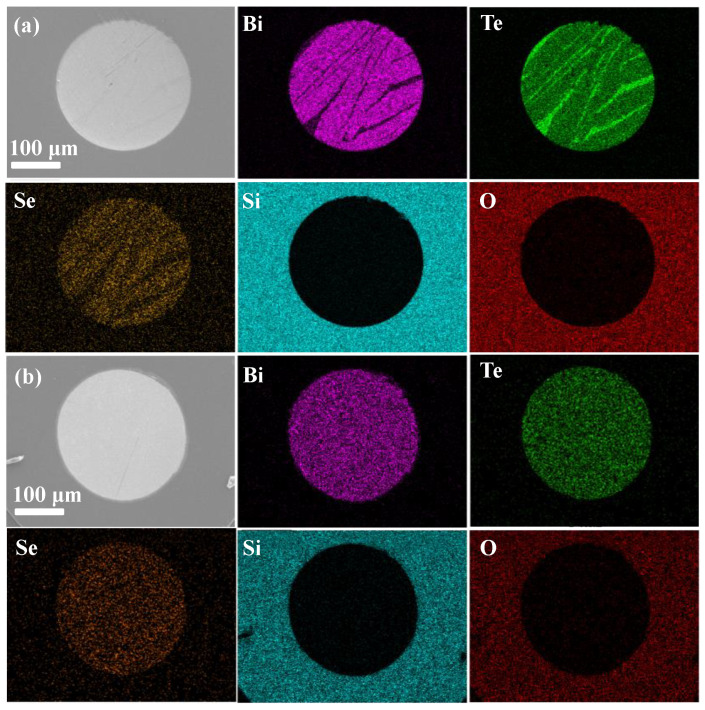
Cross-section SEM image and EDS elemental mapping of (**a**) the as-drawn and (**b**) the Bridgman annealed n-type Bi_2_Te_3_-based fiber.

**Figure 4 materials-15-05331-f004:**
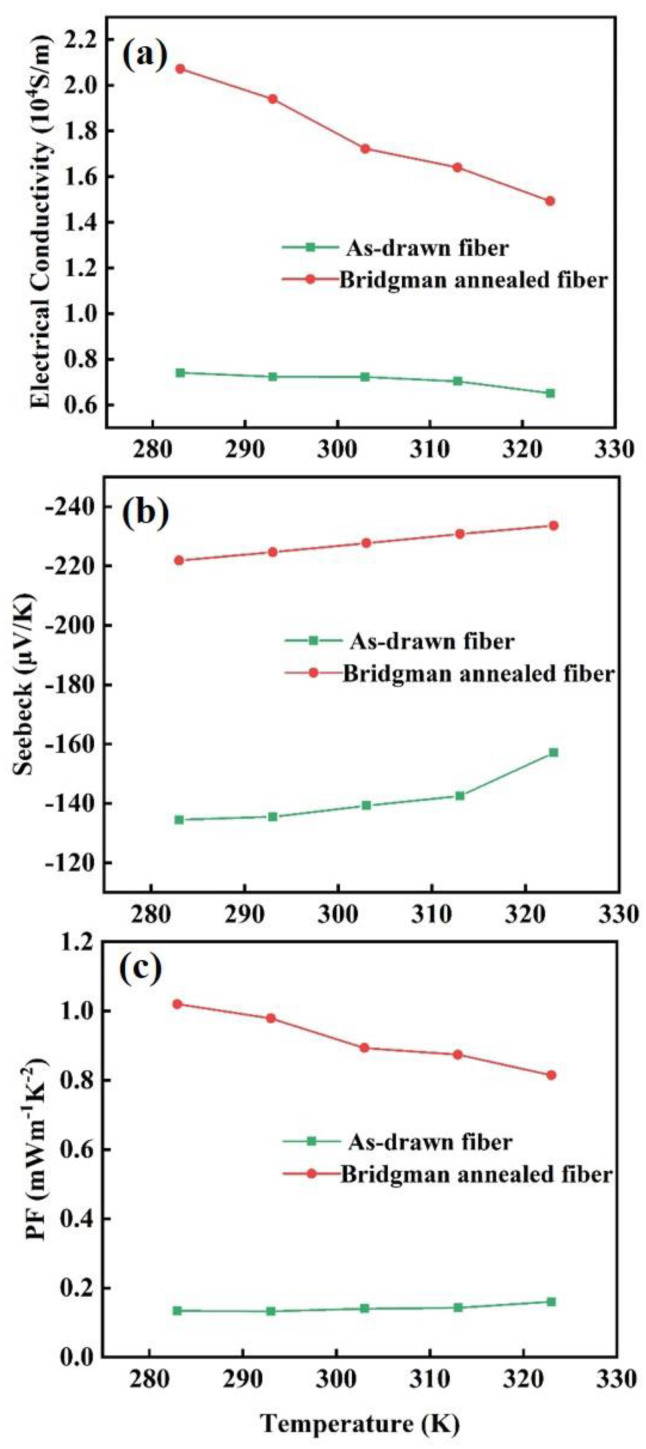
(**a**) Electrical conductivities, (**b**) Seebeck coefficients, and (**c**) power factors of the as-drawn fibers and the Bridgman-annealed fibers at 283–323 K.

**Table 1 materials-15-05331-t001:** Electrical conductivity, Seebeck coefficient, thermal conductivity, and *ZT* of the n-type Bi_2_Te_3_-based fibers at ~300 K.

Samples	Electrical Conductivity *σ* (S/cm)	Seebeck Coefficient *S* (μV/K)	Thermal Conductivity *κ*_||_ (W/mK)	*ZT*
Bridgman-annealed fiber	180 ± 7	−227 ± 11	0.64 ± 0.06	0.43
As-drawn fiber	71 ± 3	−138 ± 6	0.39 ± 0.04	0.11
Bi_2_Se_3_ fiber [16]	763 ± 35	−92 ± 6	0.84 ± 0.08	0.23
Bi_2_Se_3_ core fiber [27]	319 ± 15	−150 ± 7	1.25 ± 0.12	0.18

## Data Availability

The production data are available on request from the corresponding author.

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
