# Peer review of "Enhanced N-Type Bismuth-Telluride-Based Thermoelectric Fibers via Thermal Drawing and Bridgman Annealing"

_materials, 2022, doi:10.3390/ma15155331_

Round 1
Reviewer 1 Report
Please view the uploaded document file. Thank you.

Author Response
Reviewer #1:
This article describes the fabrication of n-type Bi2Te3-based core glass-clad fibres via thermal drawing and Bridgman annealing. At 300 K, the ZT=0.43 of the Bridgman-annealed Bi2Te3-based core is three times larger than that of the as-drawn core, and the 200-μm-diameter annealed fibres can be bent reversibly into a 2-cm curvature radius to demonstrate their flexibility. This method of drawing and annealing fibres as a proof-of-concept has promising applications in fiber-based TE conversion. However, some recommendations should be taken into account for publication.
Response: We thank the reviewer for the positive comments regarding our novel method to fabricate n-type Bi2Te3-based fibers by the thermal drawing and Bridgman annealing process. The responses to the recommendations are provided as follows.
- Introduction
In paragraph 3 at the last sentence, “The ZT value of fibers is measured to be ~0.43 at 300 K, which is approaching four times of their as-drawn fiber counterparts.” Is this your measured results? If yes, I think the authors should remove the sentence as it is redundant in the conclusion. If the sentence is the same, why in the introduction saying four times while in the conclusion 3 times?
Response: We thank the reviewer for identifying this point. Yes. “The ZT value of fibers is measured to be ~0.43 at 300 K, which is approaching four times of their as-drawn fiber counterparts”, as it can be observed at Table 1. We have revised it at the Conclusion of the revised manuscript.
- Methodology
Since the fibre is in micron size, the authors should precise in the experimental procedure. If can include a schematic draw on how the fiber were measured by the four-probe method would be nice.
Response: We agree that the experimental procedure of the four-probe method should be precise. As the similar measuring setup of the four-probe method is described in detail in our previous study, which is recently online on July 11th [Adv. Mater. 2022, 202202942], so we cite the reference in the second paragraph of the 2.2 Measurements of the revised manuscript.
- Results and Discussion
Nothing is said about the reproducibility of the measurements. How many different fibers were measured during this study?
Response: We are grateful to the reviewer for the suggestion. Indeed, we have measured three fibers for one sample and the relative deviation is lower than 5%, as shown in the 2.2 Measurements, meaning the reproducibility of the measurements is acceptable. And we have revised it in the second paragraph of the 2.2 Measurements and have added the relative deviations of electrical conductivity, Seebeck coefficient, and thermal conductivity in Table 1 of the revised manuscript.
Reviewer 2 Report
This paper reports the thermoelectric properties of Bi-Te-based thermoelectric fiber materials, and Bridgman annealing could enhance zT value. The reported sample process is interesting and promising for synthesizing fiber materials. I think the manuscript is well written and suited for publication in Materials. However, the following is my concern before judging the decision.
(1) The observed zT value for fiber materials is lower than the bulk Bi-Te-based materials. Although the thermal conductivity is significantly depressed, what is the main reason for to lower zT value?
(2) As fundamental materials research, the present paper is interesting to investigate. What is the application using fiber thermoelectric materials? The challenging issue in this research field should be addressed clearly in the introduction.
(3) Please check the data of elemental mapping for Te (green?) and Se (brown?) in Figure 3.
Author Response
Reviewer #2:
This paper reports the thermoelectric properties of Bi-Te-based thermoelectric fiber materials, and Bridgman annealing could enhance zT value. The reported sample process is interesting and promising for synthesizing fiber materials. I think the manuscript is well written and suited for publication in Materials. However, the following is my concern before judging the decision.
Response: We thank the reviewer for the publication recommendation in Materials. The responses to the several concerns raised by the reviewer are provided as follows.
- The observed zT value for fiber materials is lower than the bulk Bi-Te-based materials. Although the thermal conductivity is significantly depressed, what is the main reason for to lower zT value?
Response: Thanks for the reviewer’s constructive comment. Yes, the fabricated n-type Bi2Te3-based fibers show a lower ZT value than the best n-type nanostructured bulk [Energy & Environmental Science, 2020, 13: 2106]. The main reason should be the fiber cores show a relatively low density of ~97% and low electrical conductivity, as shown in the 2.1 Fabrication and Table 1, and our future work will be enhancing density and regulating the better component and the microstructure of the fibers. And we have added it to the Conclusion of the revised manuscript.
- As fundamental materials research, the present paper is interesting to investigate. What is the application using fiber thermoelectric materials? The challenging issue in this research field should be addressed clearly in the introduction.
Response: We thank the reviewer for raising this important point. Following the reviewer's advice, the application and the challenging issues of using fiber thermoelectric materials are added and cited at the end of the second paragraph of the Introduction of the revised manuscript:
“Although p-type Bi2Te3 fibers are reported with a high ZT~1.4 at room temperature [20], n-type Bi2Te3 fibers have not been systematically studied or improved ZT values effectively by thermal drawing or post-treatment, resulting in TE fiber devices with p-n pairs can not be widely used. The p-n pair fibers then have the potential to be applied in the field of wearable self-powered devices and temperature sensing fabrics [21-23]”
- Please check the data of elemental mapping for Te (green?) and Se (brown?) in Figure 3.
Response: Thanks for the reviewer’s careful reading. Indeed, the data of elemental mapping for Te should be green and the Se should be brown. And we have revised the Figure 3 of the revised manuscript.
Reviewer 3 Report
The paper describes the technology of obtaining n-type thermoelectric material in the form of fiber, with improved properties through the annealing process. The details of the technological process are shown clearly, as well as the procedure for characterizing the obtained material.
The paper is written correctly, without weaknesses. What is important to note is the improvement from the techniques described in the references [15-19]. In other words, a link is necessary from the end of the second paragraph to the beginning of the third paragraph in the Introduction.
Author Response
Reviewer #3:
The paper describes the technology of obtaining n-type thermoelectric material in the form of fiber, with improved properties through the annealing process. The details of the technological process are shown clearly, as well as the procedure for characterizing the obtained material.
The paper is written correctly, without weaknesses. What is important to note is the improvement from the techniques described in the references [15-19]. In other words, a link is necessary from the end of the second paragraph to the beginning of the third paragraph in the Introduction.
Response: Thanks for the reviewer’s instructive comment, which would help us to improve this manuscript. Following the reviewer's advice, the application and the challenging issues of using fiber thermoelectric materials are added and cited at the end of the second paragraph of the Introduction of the revised manuscript:
“Although p-type Bi2Te3 fibers are reported with a high ZT~1.4 at room temperature [20], n-type Bi2Te3 fibers have not been systematically studied or improved ZT values effectively by thermal drawing or post-treatment, resulting in TE fiber devices with p-n pairs can not be widely used. The p-n pair fibers then have the potential to be applied in the field of wearable self-powered devices and temperature-sensing fabrics [21-23].”
Round 2
Reviewer 1 Report
The authors have addressed all concerns.